# CiteBench: A Benchmark for Scientific Citation Text Generation

**Martin Funkquist**[1]*, **Ilia Kuznetsov**[2], **Yufang Hou**[3], **Iryna Gurevych**[2]

[1]Linköping University

[2]UKP Lab, Department of Computer Science and Hessian Center for AI (hessian.AI)
Technical University of Darmstadt

[3]IBM Research Europe - Ireland

`martin.funkquist@liu.se`

## Abstract

Science progresses by building upon the prior body of knowledge documented in scientific publications. The acceleration of research makes it hard to stay up-to-date with the recent developments and to summarize the ever-growing body of prior work. To address this, the task of citation text generation aims to produce accurate textual summaries given a set of papers-to-cite and the citing paper context. Due to otherwise rare explicit anchoring of cited documents in the citing paper, citation text generation provides an excellent opportunity to study how humans aggregate and synthesize textual knowledge from sources. Yet, existing studies are based upon widely diverging task definitions, which makes it hard to study this task systematically. To address this challenge, we propose CITEBENCH: a benchmark for citation text generation that unifies multiple diverse datasets and enables standardized evaluation of citation text generation models across task designs and domains. Using the new benchmark, we investigate the performance of multiple strong baselines, test their transferability between the datasets, and deliver new insights into the task definition and evaluation to guide future research in citation text generation. We make the code for CITEBENCH publicly available at `https://github.com/UKPLab/citebench`.

## 1 Introduction

Citations are a key characteristic of scientific communication. A paper is expected to substantiate its arguments and relate to prior work by means of exact bibliographic references embedded into the text, and the accumulated number of citations serves as a metric of publication importance. The acceleration of research and publishing in the past decades makes it increasingly challenging to both stay up-to-date with the recent publications and to

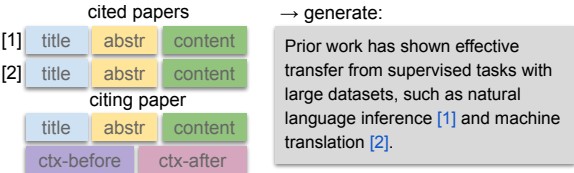

Figure 1: Citation text generation task, text in the grey box adopted from Devlin et al. (2019).

aggregate information from the ever-growing pool of pre-existing work, affecting both junior and expert researchers alike. The task of *related work generation* aims to facilitate these processes by providing topic-based multi-document summaries given a pre-defined set of papers-to-cite (Hu and Wan, 2014; Hoang and Kan, 2010).

The applications of related work generation would greatly reduce the reading and writing effort that accompanies research. Yet, the importance of related work generation spans beyond practical applications. Unlike most other genres, scientific writing enforces the use of citation markers that point to the information source, which in most cases is itself a highly structured scientific text backed by further citations. By circumventing the challenge of establishing information provenance, related work generation provides a unique opportunity to study knowledge aggregation and synthesis from textual sources, and contributes to our better understanding of text work in general.

A key sub-task of related work generation is *citation text generation* – generating citation text for pre-selected *cited* papers given the *context* of the *citing* paper (Figure 1). This task is actively studied by the NLP community and a plethora of approaches and datasets have been proposed in the past years: it has been cast as extractive (Hoang and Kan, 2010; Hu and Wan, 2014) and abstractive summarization (AbuRa'ed et al., 2020; Xing et al., 2020; Ge et al., 2021; Chen et al., 2021; Luu et al., 2021; Lu et al., 2020; Shah and Barzilay, 2021;

---

*Work done during an internship at UKP Lab.

Arita et al., 2022); taking one (AbuRa'ed et al., 2020; Xing et al., 2020; Ge et al., 2021) or multiple papers as input (Hu and Wan, 2014; Lu et al., 2020; Chen et al., 2021; Shah and Barzilay, 2021), and aiming to generate a single citation sentence (AbuRa'ed et al., 2020; Xing et al., 2020; Ge et al., 2021; Luu et al., 2021) or a paragraph (Lu et al., 2020; Chen et al., 2021).

Yet, the divergence in the existing task definitions and evaluation setups (Table 1) prevents the systematic study of the task. A common task formulation and a unified evaluation setup that would enable fair comparison of citation text generation models are missing. To address this gap, we contribute (1) CITEBENCH: a citation text generation benchmark that brings together four existing task designs by casting them into a single, general task definition, and unifying the respective datasets. We couple our benchmark with (2) a range of baseline implementations, and (3) a standardized evaluation kit complemented by additional diagnostic modules for qualitative intent- and discourse-based analysis of citation texts. We use CITEBENCH to (4) systematically investigate the task of citation text generation, revealing qualitative and quantitative differences between the existing datasets.

CITEBENCH adds substantial value to the research in citation text generation. It enables systematic comparison of existing approaches across a range of task architectures and domains, and provides a scaffolding for future citation text generation efforts. Our proposed qualitative evaluation methodology based on citation intent and discourse structure advances the state of the art in assessing the performance and properties of citation text generation models beyond shallow evaluation metrics.

## 2 Related work

### 2.1 Benchmarking

NLP benchmarks are unified dataset collections coupled with evaluation metrics and baselines that are used to systematically compare the performance of NLP systems for the targeted tasks in a standardized evaluation setup. Well-constructed benchmarks can boost progress in the corresponding research areas, such as SQuAD (Rajpurkar et al., 2016) for question answering, GLUE (Wang et al., 2018) for natural language understanding, KILT (Petroni et al., 2021) for knowledge-intensive tasks, GEM (Gehrmann et al., 2021, 2022) for general-purpose text generation, and DynaBench

(Kiela et al., 2021) for dynamic benchmark data collection. CITEBENCH is the first benchmark for the citation text generation task.

### 2.2 Text generation for scientific documents

Scientific documents are characterized by academic vocabulary and writing style, wide use of non-linguistic elements like formulae, tables and figures, as well as structural elements like abstracts and citation anchors. Recent years have seen a rise in natural language generation for scientific text, including text simplification (Luo et al., 2022), summarization (Qazvinian and Radev, 2008; Erera et al., 2019; Cachola et al., 2020), slides generation (Sun et al., 2021), table-to-text generation (Moosavi et al., 2021), and citation text generation (Li and Ouyang, 2022). Closely related to the task of citation text generation, Luu et al. (2021) study how scientific papers can relate to each other, and how these relations can be expressed in text. Related to our work, Mao et al. (2022) propose a benchmark for scientific extreme summarization. Compared to extreme summarization, which amounts to generating short context-independent summaries of individual manuscripts, citation text generation focuses on *context-dependent* descriptions that relate the cited papers to the citing paper. In line with the recent efforts that address the lack of systematic automated evaluation of natural language generation in general (Gehrmann et al., 2021), our paper contributes the first unified benchmark for citation text generation in the scientific domain.

### 2.3 Citation text generation

The task of citation text generation was introduced in Hoang and Kan (2010), who generate a summary of related work specific to the citing paper. Since then, several task definitions and setups have been proposed (Table 1). Lu et al. (2020) cast the task as generating a multi-paragraph related work section given the abstracts of the citing paper and of the cited papers. AbuRa'ed et al. (2020) use the cited paper's title and abstract to generate a citation sentence. Xing et al. (2020) use the abstract of the cited paper and include context before and after the citation sentence as the input, and produce the citation sentence as the output. A recent work by Chen et al. (2021) uses multiple cited abstracts as input to generate a related work paragraph. The great variability of the task definitions and setups in citation text generation prevents the study of ci-

| Dataset | Input | | | | | Output | | Data sources |
|---|---|---|---|---|---|---|---|---|
| | Cited document ($D^t$) | | | Citing context ($C^s$) | | Citation text ($T$) | | |
| | Single Abs | Multi Abs | Title | Abs | Text | Sent | Para | |
| ABURAED | ✓ | | ✓ | | | ✓ | | Multiple |
| CHEN | | ✓ | | | | | ✓ | S2ORC and Delve |
| LU | | ✓ | | ✓ | | | ✓ | arXiv.org and MAG |
| XING | ✓ | | | ✓ | ✓ | ✓ | | AAN |

Table 1: Overview of datasets in CITEBENCH. Single Abs = Single abstract, i.e., one cited document per instance. Multi Abs = Multiple abstracts, i.e., multiple cited documents per instance. Abs = Abstract, i.e., a dataset contains the abstract of the citing paper. Text = a dataset contains additional context from the citing paper. Sent = generation target is a single sentence. Para = generation target is a paragraph.

| Dataset | #Train | #Validation | #Test | Inputs > 4,096 tok. | Outputs > 1,024 tok. |
|---|---|---|---|---|---|
| ABURAED | 15,000 | 1,384 | 219 | 0% | 0% |
| LU | 30,369 | 5,066 | 5,093 | <0.001% | 0% |
| XING | 77,086 | 8,566 | 400 | <0.001% | <0.001% |
| CHEN | | | | | |
| − Delve | 72,927 | 3,000 | 3,000 | <0.001% | 0.004% |
| − S2ORC | 126,655 | 5,000 | 5,000 | 0.017% | <0.001% |
| **Total** | 322,037 | 23,016 | 13,712 | 0.007% | <0.001% |

Table 2: Datasets statistics. The validation set for XING has been created by us via randomly sampling 10% of the original training data. Across datasets, very few inputs contain more than 4,096 tokens, and few outputs are longer than 1,024 tokens. We exploit this property to speed up the evaluation in Section 3.3.

tation text generation methods across datasets and evaluation setups. Unlike prior work that explores *varying* task settings, CITEBENCH brings the diverging task definitions and datasets together in a *unified* setup. This allows us to compare citation text generation models across different datasets in a standardized manner using an extensive set of quantitative metrics, as well as novel automated qualitative metrics.

## 3 Benchmark

### 3.1 Task definition and datasets

We formalize the task of *citation text generation* as follows: Given a set of $n$ (cited) target documents $\{D^t_1...D^t_n\}$, a (citing) source document $D^s$ and a set of $m$ citing document contexts $\{C^s_1...C^s_m\} \in D^s$, generate a citation text $T'$ that is as close as possible to the original citation text $T \in D^s$. This general definition allows wide variation in how the task is implemented. The cited document $D^t_i$ can be represented by the abstract $a^{t_i}$, the concatenation of the title and the abstract, or even the full text of the paper. The context set $C^s$ covers sentences before and after the citation text in $D^s$, as well as the abstract $a^s \in D^s$.

Such general, open definition allows us to accommodate diverse approaches to the task within one framework (Table 1). To populate the bench-

mark, we select four datasets, focusing on the task design and domain variety: ABURAED (AbuRa'ed et al., 2020), CHEN (Chen et al., 2021), LU (Lu et al., 2020), and XING (Xing et al., 2020). Dataset transformation details are provided in Appendix A.1. Table 2 shows the quantitative statistics, and Figure 2 provides data examples from each dataset. The CHEN dataset has two subsets – CHEN Delve and CHEN S2ORC – based on the data source; we use CHEN to denote the union of the two subsets. The datasets are distributed under varying licenses; we have obtained explicit permissions from the authors to use the data for research purposes in cases when licensing was underspecified (see Ethics statement).

We note that the datasets included in the benchmark are not only structurally diverse, but also cover a wide range of domains, from medicine to computer science. In particular, ABURAED and XING exemplify citation text generation in the computational linguistics domain, CHEN Delve cover the computer science domain; LU and CHEN S2ORC span a wide range of domains represented on arxiv.org and in the S2ORC corpus, respectively, including biology, medicine and physics.

### 3.2 Evaluation and analysis kit

CITEBENCH uses two quantitative evaluation metrics to estimate the performance of the models.

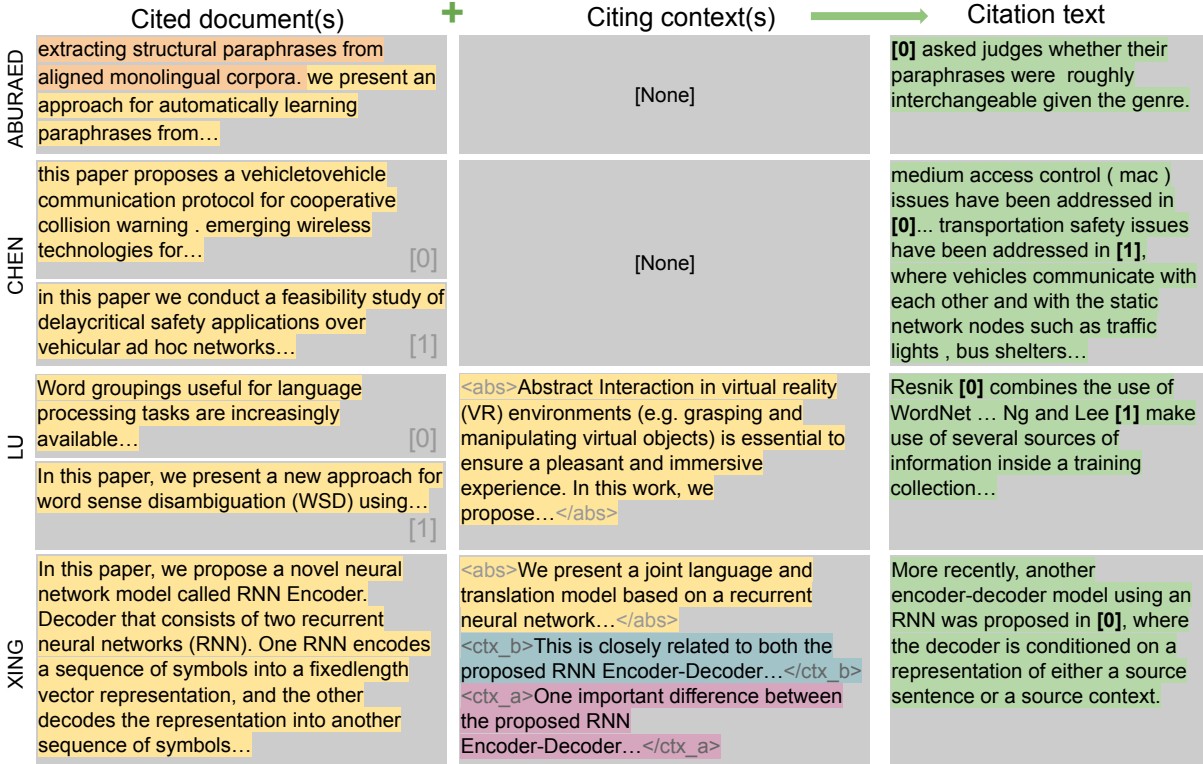

Figure 2: Data examples extracted from the datasets (left). <abs> – citing paper's abstract, <ctx_b> – context before target citation text, <ctx_a> – context after target citation text, rightmost column – generated citation text. Title in ABURAED marked with orange.

**ROUGE** (Lin, 2004) measures the n-gram overlap between the output text and the reference text. Following the recently published GEM benchmark (Gehrmann et al., 2021), we use the Huggingface ROUGE calculation package[1], with the original citing text and the model outputs as the input, without additional stemming and lemmatization. **BERTScore** (Zhang et al., 2020) is a recent alternative to ROUGE that uses contextual embeddings to calculate a similarity score between the model outputs and reference texts. We use the BERTScore implementation provided by Huggingface[2].

Quantitative metrics offer a coarse, high-level picture of comparative model performance, but deliver little *qualitative* insights into the model behavior. To alleviate this, we enrich our kit with two recently proposed citation analysis tools to study the discourse structure of citation texts – citation intent labeling (Cohan et al., 2019) and CORWA tagging (Li et al., 2022), discussed below.

**Citation intent labeling** uses the fine-grained

ACL-ARC schema by Jurgens et al. (2018) to classify citation sentences into six intents: Background provides background information; CompareOrContrast explains similarities or differences between the cited and the citing paper; Extends builds upon the cited papers; Future suggests cited papers as potential basis for future work; Motivation illustrates a research need, and Uses indicates the use of data or methods from the cited papers. **CORWA** uses the sentence classification schema proposed by Li et al. (2022), covering six citation sentence types: Single_summ and Multi_summ refer to citation sentences that are detailed summaries of a single and multiple cited papers, respectively; Narrative_cite are high-level statements related to the cited papers; Reflection sentences relate the cited paper to the current work with focus on the citing paper; Transition are non-citation sentences that connect different parts of related work, and Other accommodates all other sentences.

The two schemata offer complementary views on the citation text analysis: while citation intent focuses on *why* a paper is cited, the CORWA schema offers insights on *how* the citation text is composed

---

[1] https://huggingface.co/spaces/evaluate-metric/rouge

[2] https://huggingface.co/spaces/evaluate-metric/bertscore

and what role it plays in the surrounding text. Both schemata offer valuable insights on the composition of citation texts, and our analysis toolkit uses the publicly available model and implementation by Cohan et al. (2019)[3] for ACL-ARC-style citation intent analysis, and the publicly available code and model from Li et al. (2022)[4] for CORWA tagging. The authors report F1 scores of 67.9 for citation intent classification and 90.8 for CORWA tagging on ACL Anthology corpus (Bird et al., 2008) and S2ORC respectively, indicating that the CORWA tagger can be used with confidence for gaining insights into citation text generation, particularly on citation texts from the domains similar to ACL and S2ORC.

## 3.3 Baselines

We complement our evaluation setup with widely used **unsupervised extractive baselines** from previous work. LEAD (Lu et al., 2020) selects the first three sentences from the input, represented as the concatenation of the cited abstracts, as citation text. TextRank (Mihalcea and Tarau, 2004) and LexRank (Erkan and Radev, 2004) are graph-based unsupervised models for extractive text summarization. For TextRank, we use the default settings from the package *summa*[5]. For LexRank, we use the package *lexrank*[6] with the default settings and summary size of three sentences to match LEAD.

In addition, we provide a range of new **distantly supervised abstractive baselines** based on the pretrained Longformer Encoder Decoder (LED) (Beltagy et al., 2020). While LED is capable of handling inputs up to 16,384 tokens, to reduce the computational overhead, we truncate the inputs to 4,096 tokens, substantially reducing the computation cost while only affecting a negligible proportion of data instances (Table 2). We experiment with three different versions of the LED model: led-base, led-large and led-large-arxiv [7] which is fine-tuned on the arXiv dataset for long document summarization (Cohan et al., 2018).

Finally, CITEBENCH features a range of **directly supervised abstractive baselines** marked with *: the *led-base and *led-large-arxiv models

---

[3]https://github.com/allenai/scicite
[4]https://github.com/jacklxc/CORWA
[5]https://github.com/summanlp/textrank
[6]https://github.com/crabcamp/lexrank
[7]https://huggingface.co/allenai/led-base-16384; https://huggingface.co/allenai/led-large-16384; https://huggingface.co/allenai/led-large-16384-arxiv

---

are fine-tuned on the mixture of all training splits of all datasets in the benchmark; the *led-base model is trained for 3 epochs with batch size 16 on 4 GPUs; the *led-large-arxiv model is trained for 3 epochs with batch size 8 on 4 GPUs. In transfer learning experiments, we train and evaluate the *led-base-[X] models, each fine-tuned on a specific dataset $X$, for example, *led-base-xing.

## 4 Results

### 4.1 Baseline performance

Table 3 reports the baseline performance on the CITEBENCH datasets in ROUGE-L[8] and BERTScore. As a coarse reference point, and keeping in mind the differences in evaluation settings (Section 5.1), Lu et al. (2020) report ROUGE-L score of 30.63 for their best supervised, task-specific model, and 18.82 for LEAD. We note that our extractive LEAD baseline outperforms the distantly supervised abstractive led-base and led-large baselines on *all* datasets in terms of *all* different ROUGE metrics and BERTScore. The led-large-arxiv baseline systematically outperforms led-base and led-large as well, which we attribute to the in-domain fine-tuning on the arXiv data that consists of scientific text similar to CITEBENCH datasets. Directly supervised baselines (*led-base and *led-large-arxiv) achieve the best overall performance. We note that with few exceptions the rankings produced by the two evaluation metrics correlate: the best performing models on ROUGE also achieve the best performance on BERTScore, except *led-base that performs best on the CHEN Delve subset in terms of BERTScore but slightly falls behind *led-large-arxiv in terms of ROUGE-L.

### 4.2 Transfer learning results

The unified task formulation in CITEBENCH allows us to explore transfer learning between different domains and citation text generation setups. We examine the transfer learning performance using the *led-base-[X] models fine-tuned on individual datasets, starting from the pre-trained led-base model. Table 4 presents the results in ROUGE-L and BERTScore; Table 7 (Appendix) provides additional details. Expectedly, the models perform best both on ROUGE-L and BERTScore when evaluated on the test portion of the same

---

[8]ROUGE-1 and ROUGE-2 scores are provided in the Appendix for the sake of completeness.

| Model | ABURAED | | CHEN Delve | | CHEN S2ORC | | LU | | XING | |
|---|---|---|---|---|---|---|---|---|---|---|
| | R-L | BertS | R-L | BertS | R-L | BertS | R-L | BertS | R-L | BertS |
| LEAD | 11.32 | 75.42 | 11.48 | 74.70 | 11.34 | 74.33 | 13.34 | 75.89 | 10.55 | 75.25 |
| TextRank | 9.35 | 64.59 | 14.04 | 76.41 | 12.82 | 74.91 | 12.75 | 72.61 | 6.61 | 45.97 |
| LexRank | 10.80 | 74.90 | 12.93 | 75.96 | 12.85 | 75.11 | 14.24 | 76.70 | 10.06 | 75.14 |
| led-base | 9.06 | 74.84 | 5.55 | 70.86 | 5.41 | 70.55 | 7.36 | 72.35 | 10.07 | 74.87 |
| led-large | 8.30 | 73.26 | 6.22 | 69.57 | 6.22 | 69.77 | 6.89 | 70.51 | 9.35 | 74.30 |
| led-large-arxiv | 10.22 | 75.01 | 13.37 | 76.02 | 12.89 | 75.45 | 14.41 | 76.65 | 10.23 | 75.08 |
| *led-base | $13.44_{(0.03)}$ | 78.75 | 15.93 | **78.32** | **15.94** | **78.72** | 15.95 | 79.32 | $\textbf{13.58}_{(0.01)}$ | **78.49** |
| *led-large-arxiv | **14.90** | **79.0** | **16.27** | 78.13 | 15.92 | 78.58 | **16.53** | **79.41** | $12.42_{(0.01)}$ | 77.57 |

Table 3: Baseline performance per dataset, ROUGE-L and BERTScore, average over three runs, best scores are bolded. For readability, only standard deviation equal or higher than $10^{-2}$ is shown (in parentheses).

| Model | ABURAED | | CHEN Delve | | CHEN S2ORC | | LU | | XING | |
|---|---|---|---|---|---|---|---|---|---|---|
| | R-L | BertS | R-L | BertS | R-L | BertS | R-L | BertS | R-L | BertS |
| *led-base-aburaed | **13.95** | **78.35** | 9.96 | 75.24 | 10.66 | 75.63 | 11.63 | 76.82 | 12.63* | 78.00* |
| *led-base-chen | 11.48 | 76.57 | **16.04** | **78.23** | **16.21** | **78.71** | 15.40* | 78.58* | 10.14 | 75.38 |
| *led-base-lu | 12.19* | 76.31 | 14.58* | 77.55* | 14.59* | 77.11* | **16.25** | **79.25** | 11.56 | 77.03 |
| *led-base-xing | 11.97 | 77.56* | 9.54 | 75.06 | 10.01 | 74.57 | 9.88 | 75.66 | **14.06** | **78.70** |

Table 4: In- and cross-dataset fine-tuned model performance in ROUGE-L and BERTScore on one run (standard deviations on multiple runs are very small for all models). Best scores are bolded, * denotes the second best scores. We use the union of two subsets in CHEN for fine-tuning and evaluation.

dataset (in-domain). We note that cross-dataset (out-of-domain) transfer in several cases outperforms the strong unsupervised baselines. For instance, *led-base-chen achieves better results than all unsupervised models and off-the-shelf neural models on the ABURAED and LU datasets.

We also observe that the models often achieve better transfer-learning performance on the test sets that have task definitions similar to the training data. For instance, *led-base-aburaed and *led-base-chen reach the best out-of-domain scores on the XING dataset and the LU dataset, respectively. As indicated by Table 1, both ABURAED are XING generate a citation sentence for a single cited document, while CHEN and LU generate a citation paragraph for multiple cited papers. We elaborate on this observation in the Appendix A.4.

## 4.3 Discourse analysis

We now use the discourse analysis tools introduced in Section 3.2 to qualitatively compare the citation texts found in datasets and generated by the models. We apply the citation intent and CORWA taggers to the baseline test set outputs and macro-average the results across all datasets in CITEBENCH. We compare the resulting distributions to the distributions in the true citation text outputs found in the datasets, along with a macro-averaged total over the datasets. To quantify the discrepancies, we calculate KL divergence between the label distributions in model outputs and individual datasets.

The distribution of citation intents in Figure 3 suggests a discrepancy between the generated and original citation texts: while the baselines tend to under-generate the Background and CompareOrContrast sentences, they produce more Future, Uses and Extends sentences than the gold reference. The KL divergence of citation intent distributions (Figure 3) shows that all models' outputs are more deviated from the original citation intentions on ABURAED compared to other datasets. Interestingly, the two fine-tuned models that perform well in terms of ROUGE scores tend to also achieve lower KL divergence scores for all datasets, e.g., *led-base and *led-large-arxiv. This suggests that the two models learn a dataset's citation intent distribution during fine-tuning.

Turning to the CORWA results, Figure 4 suggests high discrepancy between the baseline model ouputs and the test set outputs. Most of our baselines under-generate the Narrative_cite and – surprisingly – Single_summ class, while over-generating the Reflection, compared to the distributions in the gold reference texts. The only two exceptions are the fine-tuned *led-base and *led-large-arxiv models, which aligns with their high performance in terms of ROUGE and BERTScore. High KL divergence values in Figure 4 confirm that the predicted CORWA tags are more discriminative across the datasets compared to ci-

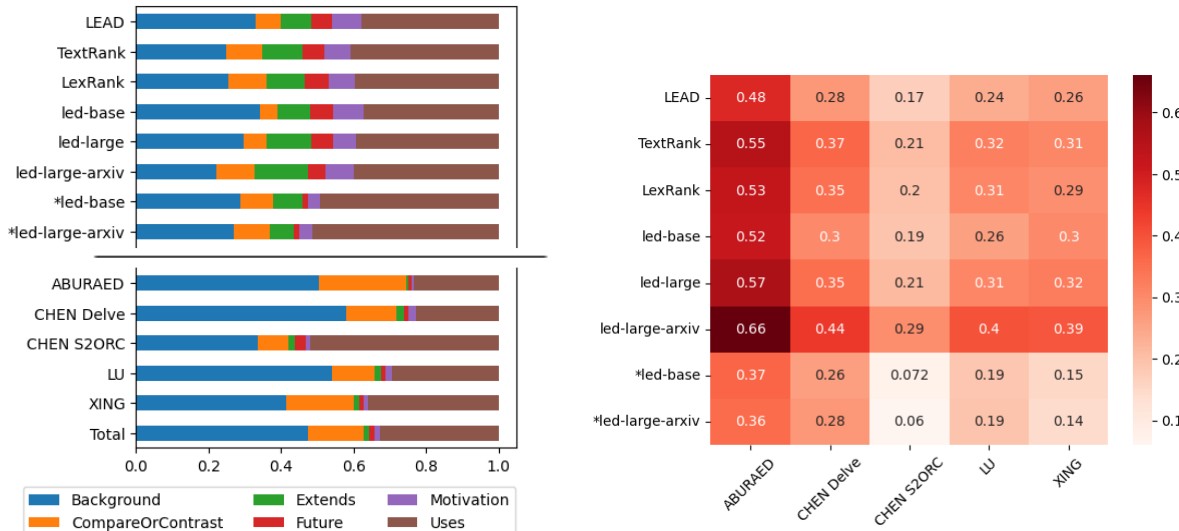

Figure 3: Citation intent distribution (left) for model outputs (top) and datasets (bottom), and KL divergence between datasets and model outputs (right).

tation intent labels. The lowest CORWA tag distribution divergence is observed for the `*led-base` and `*led-large-arxiv` baselines, suggesting that the learned ability to capture the CORWA tag distribution might play a role in improved citation text generation performance.

## 5 Discussion

### 5.1 Replicability of ROUGE

While re-implementing the baselines, we found that replacing the Huggingface ROUGE implementation with the *files2rouge* package used by Lu et al. (2020) results in a substantial ROUGE-1/L score increase of approx. 3 points. Our investigation revealed that this discrepancy is due to preprocessing: both packages apply transformations to the inputs, yet, while *files2rouge* includes stemming by default, the Huggingface implementation does not. We found additional effects due to tokenization and stopword removal. This implies that the differences in ROUGE scores across publications can be attributed to the particularities of the evaluation library, and not to the merits of a particular model. This underlines the importance of clearly specifying the ROUGE package and configuration in future evaluations.

### 5.2 Qualitative evaluation

None of the existing automatic evaluation metrics for text generation can substitute in-depth qualitative analysis of model outputs by humans. Yet,

such an analysis is often prohibitively expensive, limiting the number of instances that can be manually assessed. The progress in automatic discourse analysis of citation texts makes it possible to study the composition of reference texts and compare it to the aggregate model outputs. We proposed citation intent classification and CORWA tagging as an inexpensive middle-ground solution for qualitative evaluation of citation texts. Our results indicate that the composition of citation texts indeed differs, among datasets and system outputs.

Driven by this, we used KL divergence between reference and generated texts to study the differences between citation texts in aggregate. Following up on our analysis in Section 4.3, we observe that while the two best-performing baseline models in terms of ROUGE and BERTScore also achieve the lowest KL divergence on both discourse schemata, the correlation is not perfect. We thereby suggest the use of KL divergence of citation intent and CORWA tag distributions as a supplementary metric to track progress in citation text generation, while keeping in mind that the performance of the underlying discourse analysis models is itself subject to future improvement (see Limitations).

### 5.3 Human evaluation

To get a better understanding of the model outputs, three human annotators have manually inspected 25 baseline outputs from each of the best-performing LexRank, `led-large-arxiv` and `*led-large-arxiv` models, in each model type

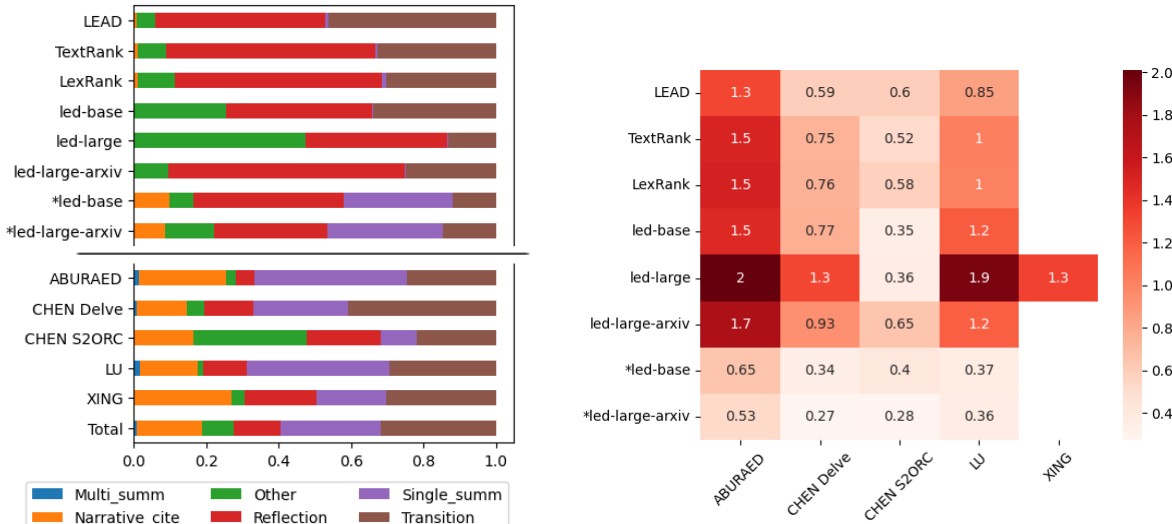

Figure 4: CORWA tag distribution (left) for model outputs (top) and datasets (bottom), and KL divergence between datasets and model outputs (right). Empty cells in XING (right) denote ∞ due to missing labels in system predictions. Note the scale differences between the KL divergence plots here and in Figure 3, kept for presentation clarity.

category, on each of the two architecturally distinct XING and CHEN DELVE datasets, 150 instances in total. Each output was rated on a 5-point scale in terms of *readability* and *consistency*. The estimated Quadratic Cohen's kappa agreement between annotators on a 50-instance subsample was approx. 0.50 for readability and ranging between 0.18 and 0.42 for consistency, depending on the annotator pair (Appendix A.5). Turning to the evaluation results, on CHEN DELVE, the fine-tuned *led-large-arxiv receives the highest average readability score of 4.32 and consistency score of 2.36, while the baseline led-large-arxiv achieves the best scores on both dimensions on XING, with an average readability score of 3.88 and consistency score of 2.20. Overall, we observe that while some models achieved satisfactory readability scores on some datasets, *none* scored higher than 3 on the consistency scale: while the generated citation texts can be topically related to the gold reference, in most cases they miss key factual details about the cited papers and the contextual information in the reference. Motivated by this, we conclude our discussion by turning to the question on the optimal definition for the citation text generation task.

### 5.4 Task refinement

Unifying a wide range of citation text generation setups within a common framework allows additional insights into the definition of the task. While CiteBench accommodates different *structural* versions of the citation text generation task, input and output structure and granularity are not the only parameters that influence task complexity, and future work in citation text generation might explore other, *qualitative* aspects of the task. A manual review of the datasets included in CITEBENCH reveals that in some cases citation text can not be generated based on the provided inputs due to missing information (see Appendix A.6 for examples). This raises the question of *what information is in fact required to produce accurate citation texts*.

In addition, we observed that prior data includes instances of self-reference where citation text in fact talks about the citing paper itself. To further substantiate this observation, we have searched citation texts throughout CITEBENCH for keywords that might indicate self-reference ("*our work*", "*in this paper*", "*we present*", "*we introduce*") and found that the datasets widely differ in terms of self-reference, from ~ 1% in the single-sentence citation text datasets (ABURAED and XING) to up to 14% in CHEN Delve (Appendix A.6). This suggests that *task design can influence the qualitative composition of citation texts*, and calls for further investigation of data selection procedures for citation text generation.

Finally, we note that citation text might vary depending on the author's intent. As our analysis in Figures 3 and 4 demonstrates, datasets do differ in terms of the distribution of citation intents and discourse tags. Lauscher et al. (2022) show that

intent classification can benefit from citation contexts, and we argue that modeling citation intent *explicitly* could lead to a more robust and realistic definition of the citation text generation task: a `Comparison` citation would differ in content and style from a `Background`. Thus, we deem it promising to extend the input with the information about intent, guiding the model towards more plausible outputs.

## 6  Conclusion

Citation text generation is a key task in scholarly document processing – yet prior work has been scattered across varying task definitions and evaluation setups. We introduced CITEBENCH: a benchmark for citation text generation that unifies four diverse task designs under a general definition, harmonizes the respective datasets, provides multiple baselines and a standard evaluation framework, and thereby enables a systematic study of citation text generation.

Our analysis delivered new insights about the baseline performance and the discourse composition of the datasets and model outputs. The baseline results show that simple extractive summarization models like `LexRank` perform surprisingly well. Furthermore, we observe non-trivial ability for transfer learning among the baselines. The discourse analysis-based evaluation suggests that the models performing best in terms of ROUGE and BERTScores capture the natural citation intent distribution in the generated texts.

Finally, our discussion outlines recommendations and promising directions for future work, which include detailed reporting of evaluation packages, the use of discourse-based qualitative metrics for evaluation, and further refinements to the citation text generation task definition. We argue that the current definitions for citation text generation overly simplify the complex process of composing a related work section, which hinders the development of successful real-world applications. In general, we view our work represents a crucial first step in addressing the extremely challenging problem of AI-assisted writing related work. We invite the community to build upon CITEBENCH by extending it with new datasets, trained models and metrics.

## Limitations

The datasets in CITEBENCH are in English, limiting our baseline models and results to English. While CITEBENCH covers a wide variety of scientific domains, humanities and social sciences are under-represented. Since the citation patterns might differ across research communities, this might lead to performance degradation due to domain shift. The over-focus on English data from natural and computer sciences is a general feature of scholarly NLP. We hope that the future developments in the field will enable cross-domain and cross-lingual applications of NLP to citation text generation.

The analysis results on citation intent and CORWA tagging are based on the output of the corresponding discourse tagging models. The performance of these models is not perfect. For citation intent classification, the best model by Cohan et al. (2019) on ACL-ARC has an F1 score of 67.9. In a further analysis of the classification errors, Cohan et al. (2019) note that the model tends to make false positive errors for the `Background` class and that it confuses `Uses` with `Background`. For CORWA tagging, Li et al. (2022) report an F1 score of 90.8 for their best-performing model. The limitations of automatically inferred labels need to be taken into consideration when interpreting the results. Given the data source overlap between the CORWA and ACL-ARC datasets and CITEBENCH, we have attempted to find overlapping instances to estimate the performance of the models on the subset of data used in CITEBENCH. Yet, the comparison between CORWA and ACL-ARC test sets and CITEBENCH test sets[9] only yielded eight overlapping instances in the LU dataset out of 14,459 total instances, preventing the further investigation. We leave the validation of our findings with human-annotated citation intents and discourse tags to future work. Furthermore, while we focus on sentences as an easy-to-obtain coarse-grained unit for discourse tagging, optimal granularity for citation analysis is another open research question to be investigated (Li et al., 2022).

While our general task definition allows incorporating any information from the target and source documents, we offer no *standardized* way to in-

---

clude structured information like citation graph context and paper metadata. Yet such information might be useful: for example, Wang et al. (2022) explore the use of citation graphs for citation text generation. We leave the standardization of additional information sources for citation text generation to the future work. Our baseline implementations limit the input sequences to 4,096 tokens, which only affects a small portion of the data. This restriction can be lifted as long as the target language model can efficiently process long documents, and experimental time is not a concern – even in a limited setting, performing the full run of all *fine-tuned* citation text generation models in the benchmark is computationally expensive (Appendix A.2). Finally, CITEBENCH inherits the structural limitations of the datasets it subsumes, e.g. not preserving the full document information and document structure, and filtering out multimodal content. We leave the investigation of these extensions to the future.

## Ethics Statement

Citation text generation is intended to support scholars in performing high-quality research, coping with information overload, and improving writing skills. Although we do not foresee negative societal impact of the citation text generation task or the accompanying data per se, we point at the general challenges related to factuality and bias in machine-generated texts, and call the potential users and developers of citation text generation applications to exert caution and to follow the up-to-date ethical policies, incl. non-violation of intellectual property rights, maintaining integrity and accountability.

In addition, the raising ethical standards in the field put new demands on clear dataset licensing. While enforcing the license on third-party datasets presents an organizational and legal challenge and lies beyond the scope of our work, we made the effort to clarify the copyright status of the existing datasets to the best of our ability. LU[10] is distributed under the open MIT license[11]. CHEN[12] openly distribute their annotations for non-commercial use, while referring to the source datasets for license on the included research publications: the S2ORC

corpus is released under the open non-commercial CC BY-NC 4.0 license[13]; yet, we could not establish the licensing status of the Delve corpus based on available sources. XING[14] is based on the openly licensed ACL Anthology Network corpus but does not specify license for the annotations. We contacted the authors to obtain explicit permission to use their annotations for research purposes. ABURAED[15] do not attach license to the data, which is partly extracted from the ACL Anthology Network Corpus. We collected an explicit permission from the authors to use their dataset for research purposes. Our observations underline the urgent need for clear standartized data licensing practices in NLP. To work around the potential limitations on data reuse, CITEBENCH provides the instructions and a script to fetch and transform the data from the respective datasets. All datasets used in this work were created for the task of citation text generation and are employed according to their intended use.

## Acknowledgements

This study is part of the InterText initiative at UKP Lab[16]. It was funded by the German Federal Ministry of Education and Research and the Hessian Ministry of Higher Education, Research, Science and the Arts within their joint support of the National Research Center for Applied Cybersecurity ATHENE; by the German Research Foundation (DFG) as part of the PEER project (grant GU 798/28-1); and by the European Union as part of the InterText ERC project (101054961). Views and opinions expressed here are, however, those of the author(s) only, and do not necessarily reflect those of the European Union or the European Research Council. Neither the European Union nor the granting authority can be held responsible for them. The work was partially supported by the Wallenberg AI, Autonomous Systems and Software Program (WASP) funded by the Knut and Alice Wallenberg Foundation.

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

# A Appendix

## A.1 Data transformation

Prior work not only explores a wide range of task definitions for citation text genration, but also employs a multitude of formats to serialize the task information in the data. Thus, to construct a unified benchmark, we also needed to unify the data. The data format used in CITEBENCH is a concatenation of different types of *inputs* embedded into tags – special tokens that denote which input is stored within. The abstracts of the cited papers are always available and are not embedded into a tag. The final CITEBENCH input is assembled as follows, depending on the availability of the inputs in the source dataset's task definition:

- <abs></abs> the abstract of the *citing* paper
- <ctx_b></ctx_b> the sentence(s) **b**efore the *citation text* in the *citing* paper
- the abstract(s) of the *cited* paper(s) discussed in the citation text (without a surrounding tag)
- <ctx_a></ctx_a> the sentence(s) **a**fter the *citation text* in the *citing* paper

The CITEBENCH data stores these input components as a list. To pass the data to the model, we simply concatenate the items on the list. In addition, for each instance we store the *gold reference* target text, which is used as output example at the training stage, and as an evaluation target at the testing stage. Table 5 provides an example of an instance coded according to the CITEBENCH schema.

Unification required us to process the original data. No instance-level filtering was applied, i.e. every instance from the source datasets is contained in the benchmark. All changes to the underlying datasets are documented in the automatic scripts supplied with this paper, which can be used to reconstruct CITEBENCH from the sources. The paragraphs below briefly outline the main modifications that these scripts perform.

For *all* datasets, we unify the citation anchor format. In CITEBENCH, cited papers in the citation text are represented by an integer placeholder (e.g. [0], [1]) which increments with each new encountered paper in the input text, but is consistent throughout the text. For example, if a paper "[0]" was encountered twice, this would result in a citation text going as "*As it has been shown in [0], [1], [2]... In particular, [0] demonstrate that...*". In XING the citations of papers that are not included in the input are marked in the original data using a

| Input | Target |
|---|---|
| ["<abs> The SRI Core Language Engine (CLE) is a general-purpose natural anguage front end for interactive systems. It translates English <...> is described and evaluated. </abs>", "<ctx_b> The system also facilitates the process of adding word forms to the user is own dictionary. <...> how to accomplish this task with the user is assisstance. </ctx_b>", "Spelling-checkers have become an integral part of most text processing software. <...> a special method has been developed for easy word classification.", "<ctx_a> The hnplementation f the algoritlma <...> it should be included in the second version). </ctx_a>"] | The idea is similar to Finkler and Neumann #OTHEREFR, though simplified for our purposes; [0] in his VEX system also uses the method of giving sunple questions to the user (supposedly non-linguist) to learn about word is behaviour, but it is for English and primarily intended for assigning syntax properties rather than morphological. |

Table 5: Example CITEBENCH instance taken from the XING dataset.

placeholder #OTHEREFR; these tokens are kept in the benchmark.

ABURAED dataset operates with single cited documents, providing their abstract and title, and aims to generate a single citation sentence. We use the script from the original study[17] to extract the input and the gold reference. The gold reference is used directly to populate the "target" field. The input is constructed by concatenating the title and the abstract of the cited paper.

Both CHEN datasets operate with multiple cited documents, storing their abstracts, and have citation paragraph as the target granularity. CHEN uses a field named "abs" to denote the citation text to generate, which is not to be confused with the "<abs></abs>" tag used in CITEBENCH to denote the citing paper abstract. This field is used as gold reference. To construct the input, we extract the "multi_doc" field from CHEN data. The abstracts of the individual documents in the source data begin with a special marker "[x]". We remove this marker and concatenate the inputs, maintaining the original order of the abstracts.

LU also follows a multiple-abstract citation text paragraph generation setting. We take the "related_work" field from the original data and use it as our golden reference "target". From the "ref_abstract" dictionary the citation markers and cited paper abstracts are extracted. The citation markers are converted to indexes e.g. "[0]" and the abstracts are ordered according to these indices. The "abstract" (citing paper's abstract) and the cited paper's abstracts from "ref_abstract" are concatenated into the model input according to the procedure described above.

Lastly, for XING the "explicit_citation" is the

gold reference and it is copied over to the "target" field in CITEBENCH, representing the gold reference. The "tgt_abstract" (cited paper's abstract) is wrapped in "<abs></abs>" tags, the "text_before_explicit_citation" (sentences before citation text) is wrapped in "<ctx_b></ctx_b>" tags, "text_after_explicit_citation" (sentences after citation text) is wrapped in "<ctx_a></ctx_a>" tags. These items are added to the "input" list together with the "src_abstract" (cited paper's abstract), using the tags and order described above.

## A.2 Model training

For the LED models, the following hyperparameters were used: Encoder length $4,096$, Decoder length $1,024$, Decoding type beam search, 2 beams, length penalty 2, with early stopping, no repeat n-gram size 3, learning rate $5e-5$. To make the study feasible given the number of experiments in this work, no hyperparameter search was performed. The *led-base and *led-large-arxiv model were fine-tuned for 9 days on 4 GPUs; each of the models fine-tuned on individual datasets *led-base-X where trained for a maximum of 3 days on 4 GPUs (most of them took less than 3 days to train). Using the upper bounds that would result in $(9d + 9d + 4 \times 3d) \times 4$ GPUs $\times 24$hours = 2880 GPU hours in total for model fine-tuning.

## A.3 ROUGE calculation

The ROUGE scores are calculated using the Huggingface *evaluate* library[18] with package versions: *rouge-score==0.0.4* and *evaluate==0.1.2*. The scores that are reported are *fmeasure* for the *mid* scores (see the library documentation for details). See Tables 6 and 7 for detailed ROUGE-L, ROUGE-1 and ROUGE-2 results.

---

[17]https://github.com/AhmedAbuRaed/SPSeq2Seq/blob/master/preprocesstarget.py

[18]https://huggingface.co/spaces/evaluate-metric/rouge

| Model | ABURAED | CHEN Delve | CHEN S2ORC | LU | XING |
|---|---|---|---|---|---|
| LEAD | 16.69 / 2.22 / 11.32 | 19.75 / 2.23 / 11.48 | 18.64 / 2.32 / 11.34 | 22.99 / 3.40 / 13.34 | 14.98 / **2.22** / 10.55 |
| TextRank | 13.59 / 1.54 / 9.35 | 27.39/3.77/14.04 | 23.09/3.48/12.82 | 23.05 / 3.98 / 12.75 | 8.51 / 1.14 / 6.61 |
| LexRank | 15.90 / 2.00 / 10.80 | 23.79 / 3.12 / 12.93 | 22.35 / 3.10 / 12.85 | 25.53 / 4.18 / 14.24 | 14.47 / 1.98 / 10.06 |
| led-base | 11.06 / 1.72 / 9.06 | 7.16 / 0.74 / 5.55 | 6.68 / 0.75 / 5.41 | 9.73 / 1.35 / 7.36 | 12.36 / 1.66 / 10.07 |
| led-large | 10.39 / 1.36 / 8.30 | 7.85 / 0.96 / 6.22 | 7.56 / 0.99 / 6.22 | 8.84 / 1.25/6.89 | 11.61/1.29/9.35 |
| led-large-arxiv | 15.55 / 1.89 / 10.22 | 25.78 / 3.35 / 13.37 | 23.45 / 3.20 / 12.89 | 26.27 / 4.44 / 14.41 | 14.57 / 2.04 / 10.23 |
| *led-base | 19.65 / 2.12 / 13.44 | 30.66 / 5.51 / 15.93 | 26.58 / 5.82 / **15.94** | 28.76 / 5.03 / 15.95 | **18.88** / 2.04 / **13.58** |
| | (0.018) / (0.006) /(0.025) | (0.004) / - / (0.002) | (0.003) / (0.003) / - | (0.007) / (0.002) / (0.003) | (0.009) / - / (0.006) |
| *led-large-arxiv | **21.56 / 2.96 / 14.90** | **31.65 / 6.01 / 16.27** | **26.70 / 6.07** / 15.92 | **30.54 / 5.63 / 16.53** | 17.90 / 1.52/ 12.42 |
| | (0.003) / (0.005) / (0.011) | (0.007) / (0.002) / (0.003) | (0.007) / (0.001) / (0.002) | (0.010) / (0.004) / (0.002) | (0.012) / (0.009) / (0.008) |

Table 6: Detailed baseline performance, ROUGE-1/2/L, average over three runs, best scores are bolded. For readability, only standard deviation higher than $10^{-4}$ is shown (in parentheses).

## A.4 Transfer Learning Results

A possible explanation for the better transfer performance between ABURAED and XING (discussed in Section 4.2 and seen in Table 4 and 7) might lie in the behavior of the ROUGE metric: as observed by Sun et al. (2019), ROUGE tends to award higher scores to texts that are around 100 tokens long. The average lengths of the outputs for ABURAED and XING are 47.5 and 38.3 words respectively. For CHEN Delve, CHEN S2ORC and LU the average output lengths are 265.8, 186.3 and 140.2 words respectively. For the inputs the average lengths for ABURAED and XING are 65.3 and 145.9 words respectively, while for CHEN Delve, CHEN S2ORC and LU the lengths are 691.4, 1230.1 and 676.1 words respectively.

## A.5 Human evaluation details

For the human evaluation we focus on the datasets XING and CHEN DELVE as these are the datasets with most divergent task definitions, and the models LexRank, led-large-arxiv and *led-large-arxiv as these are the systems with highest performance in their respective baseline categories. For each dataset, 25 data instances were randomly selected and for each of these instances the output from each model was generated. This resulted in $2 \times 3 \times 25 = 150$ instances to annotate. The instances were shuffled and distributed among three annotators (paper authors), fluent non-native English speakers with background in computer science and natural language processing, who were asked to rate each instance according to its *readability* and *consistency* with the gold reference. For both dimensions we have used a 5-point scale. Results for the human evaluation are shown in Table 8. Readability scores were defined as:

- 1: the text lacks basic grammatical coherency and is not readable;
- 2: the text lacks basic grammatical coherency but is readable with some effort;

- 3: the text is readable but has some major grammatical or fluency issues;
- 4: the text is readable and has only one or two minor grammatical or fluency issues;
- 5: the text is perfectly readable.

Consistency scores were defined as:

- 1: the text is not related to the gold reference;
- 2: the text is topically related to the gold reference but misses all key aspects and the corresponding factual details about the cited papers;
- 3: the text is topically related to the gold reference but misses or misrepresents a few key factual details about the cited papers;
- 4: the text is topically related to the gold reference and captures all key aspects about the cited papers, but it misses some factual details;
- 5: the text conveys the same message and detail as the original text.

To estimate the agreement between the annotators, we randomly selected 25 instances from each of the two annotators A and B, that were relabeled by the annotator C, resulting in a total of 50 instances. The Quadratic Cohen's kappa for the annotator pair (A-C) was measured at 0.48 for readability and 0.18 for consistency; the annotator pair (B-C) yielded the agreement of 0.52 for readability and 0.42 for consistency. We note the discrepancy with regard to the consistency score between annotator pairs[19], which indicates that compared to readability, consistency is a less intuitive notion and might require a more strict definition or training in future human evaluations.

## A.6 Discussion examples

In Table 9 two examples from the benchmark are shown where the input is not sufficient to generate the output. It is not reasonable to expect a model

---

[19] Upon close examination, annotator C tends to be stricter in evaluating the consistency score compared to annotator A.

| Model | ABURAED | CHEN Delve | CHEN S2ORC | LU | XING |
|---|---|---|---|---|---|
| *led-base-aburaed | **19.65** / 1.95 / **13.95** | 15.32 / 1.96 / 9.96 | 15.96 / 2.18 / 10.66 | 17.84 / 2.54 / 11.63 | 18.68 / 1.46 / 12.63 |
| *led-base-chen | 17.66 / **2.41** / 11.48 | **30.82** / **5.58** / **16.04** | **27.28** / **5.89** / **16.21** | 28.92 / 4.80 / 15.40 | 14.51 / 1.89 / 10.14 |
| *led-base-lu | 17.78 / 2.32 / 12.19 | 26.97 / 3.97 / 14.58 | 25.90 / 3.72 / 14.59 | **28.94** / **5.00** / **16.25** | 17.50 / 1.92 / 11.56 |
| *led-base-xing | 16.56 / 1.32 / 11.97 | 16.01 / 1.87 / 9.54 | 16.09 / 1.89 / 10.01 | 14.87 / 1.81 / 9.88 | **19.77** / **2.80** / **14.06** |

Table 7: Detailed transfer learning performance, in- and cross-dataset, ROUGE-1/2/L.

| Dataset | Model | Readability | Consistency |
|---|---|---|---|
| All | All | $3.73 \pm 1.19$ | $2.04 \pm 0.84$ |
| All | LexRank | $3.42 \pm 1.09$ | $2.04 \pm 0.75$ |
| All | led-large-arxiv | $\mathbf{3.94} \pm 1.00$ | $\mathbf{2.18} \pm 0.85$ |
| All | *led-large-arxiv | $3.90 \pm 1.39$ | $1.94 \pm 0.91$ |
| XING | All | $3.64 \pm 1.30$ | $1.92 \pm 0.85$ |
| CHEN DELVE | All | $3.87 \pm 1.06$ | $2.19 \pm 0.82$ |
| XING | LexRank | $3.56 \pm 1.12$ | $2.04 \pm 0.79$ |
| XING | led-large-arxiv | $\mathbf{3.88} \pm 1.01$ | $\mathbf{2.20} \pm 0.87$ |
| XING | *led-large-arxiv | $3.48 \pm 1.69$ | $1.52 \pm 0.77$ |
| CHEN DELVE | LexRank | $3.28 \pm 1.06$ | $2.04 \pm 0.73$ |
| CHEN DELVE | led-large-arxiv | $4.00 \pm 1.00$ | $2.16 \pm 0.85$ |
| CHEN DELVE | *led-large-arxiv | $\mathbf{4.32} \pm 0.85$ | $\mathbf{2.36} \pm 0.86$ |

Table 8: Human evaluation results; average ± standard deviation.

to be able to generate these data instances. Table
10 provides examples of self-reference, and Table
11 provides self-reference statistics.

| Input | Target | Comment |
|---|---|---|
| We present the syntax-based string-to-tree statistical machine translation systems built for the WMT 2013 shared translation task. Systems were developed for four language pairs. We report on adapting parameters, targeted reduction of the tuning set, and post-evaluation experiments on rule binarization and preventing dropping of verbs. | It is worth noting that the German parse trees [0] tend to be broader and shallower than those for English. | From this input we can not determine whether German parse trees are broader and shallower than English parse trees. |
| Data selection is an effective approach to domain adaptation in statistical machine translation. The idea is to use language models trained on small in-domain text to select similar sentences from large general-domain corpora, which are then incorporated into the training data. Substantial gains have been demonstrated in previous works, which employ standard ngram language models. Here, we explore the use of neural language models for data selection. We hypothesize that the continuous vector representation of words in neural language models makes them more effective than n-grams for modeling unknown word contexts, which are prevalent in general-domain text. In a comprehensive evaluation of 4 language pairs (English to German, French, Russian, Spanish), we found that neural language models are indeed viable tools for data selection: while the improvements are varied (i.e. 0.1 to 1.7 gains in BLEU), they are fast to train on small in-domain data and can sometimes substantially outperform conventional n-grams. | Analyses have shown that this augmented data can lead to better statistical estimation or word coverage [0]. | Here we do not know what "this" refers to in the target and the input does not mention anything about "better statistical estimation" or "word coverage" |

Table 9: Examples of targets that contain information missing from the input.

| Samples of self reference sentences in the reference citation text |
|---|
| (1) *In our context, we call upon computer vision techniques to study the cephaloocular behavior of drivers.* |
| (2) *The main objective of our work is the elaboration of a new computer vision system for evaluating and improving driving skills of older drivers (age between 65 and 80 years).* |
| (3) *The features used in this work are complex and difficult to interpret and it isn't clear that this complexity is required.* |
| (4) *Our approach enables easy pruning of the RNN decoder equipped with visual attention, whereby the best number of weights to prune in each layer is automatically determined.* |

Table 10: Self-reference sentence examples; (1) and (2) from Chen et al. (2021); (3) and (4) from Lu et al. (2020).

| Dataset | Data points | Ref to citing paper | Prevalence |
|---|---|---|---|
| ABURAED | 16,603 | 143 | 0.86% |
| CHEN Delve | 78,927 | 11,787 | 14.93% |
| CHEN S2ORC | 136,655 | 6,711 | 4.91% |
| LU | 40,528 | 4,470 | 11.03% |
| XING | 86,052 | 1,475 | 1.71% |
| **Total** | 530,869 | 28,167 | 5.31% |

Table 11: Self-reference sentence statistics.