# OpenReview forum: "CiteBench: A Benchmark for Scientific Citation Text Generation"
_EMNLP/2023/Conference — EMNLP 2023 Main_

### Official Review · Reviewer_Z3w2 · 2023-07-21

**Typos Grammar Style And Presentation Improvements:** 542
**Soundness:** 4

**Excitement:**

4: Strong: This paper deepens the understanding of some phenomenon or lowers the barriers to an existing research direction.

**Paper Topic And Main Contributions:**

This paper contributes a large benchmark of citation text generation unifying four structurally different previous datasets (see Figure 2). The authors provide an evaluation and analysis kit for both quantitative and qualitative evaluation on this task including a novel measure on discursive behavior of citation text generation mechanisms based on KL divergence between source corpora and generated text intent and composition classification. The authors provide a comprehensive set of baselines for future community members to build on as well as a number of suggestions of future directions that can be explored using this benchmark.

**Reasons To Accept:**

This is an insightful, well-written, and well-designed benchmark. The work positions itself well and responds to other works investigating this task by solving a number of problems for the community: (1) how to unify structurally different citation text generation datasets? (2) how to evaluate citation text generation (3) what are some of the main directions we should be considering in this task? (Section 5.4) The resulting benchmark, evaluations, and baselines will be valuable to the community working on this task. The baselines are thorough including performance evaluations on indirect supervision and OOD settings and a human evaluation study. The evaluations also include insightful discussions about issues with ROUGE (5.1) as well as a novel KL divergence metric which already reveals in this study a number of interesting problems for future studies to follow up on.

In addition to proposing the above, the authors provide a number of interesting insights like Section 4.3, 382-385, Figure 3 + 4: Insights about differences between generative models and gold reference discursive roles. Insights about what information is required for generating citations (Appendix A.2) and the presence of self-reference in citation generation datasets (Appendix A.2). The limitations and ethics statements also include important information for people considering working on this task.


**Reasons To Reject:**

This paper has very few weaknesses. One minor weakness I have identified is: (1) There was no validation done to check if the Scicite or CORWA labeling of either the benchmark or the generated texts actually reflect how humans would label them. SciCite and CORWA are S2ORC based, so I do think the authors could do some automatic validation to check the KL divergence between samples which have gold labels as reference points for the readers.

**Reproducibility:**

5: Could easily reproduce the results.

**Reviewer Confidence:**

5: Positive that my evaluation is correct. I read the paper very carefully and I am very familiar with related work.

---

> ### Author Rebuttal · Authors · 2023-08-28
>
> Thank you for taking your time and thoroughly reviewing our work, we appreciate your detailed feedback and respond to it below, starting with minor points.
>
> We will mention the lack of humanities and the limited citation style diversity as a data limitation, and restate the insights on the baseline performance and discourse composition in the Conclusion. We will also add information on the number of epochs. We will fix the typographical error in Table 2, thanks for spotting it! Correct values are 0.006% for the total inputs with more than 4096 tokens; the reason that the total outputs over 1024 tokens is 0% is because we round any value below 0.001 to zero. We only report RL scores in the main tables to save space. We will explicitly redirect the reader to the appendix that also provides R1 and R2 ROUGE scores. Regarding the intents (lines 529-531), what we suggest is explicitly providing intent as part of the concatenated model input. This means that it would not be an auxiliary task but rather an extension to the input of the current task. Alternatives are of course possible! We will clarify this in the text.
>
>
> -- **Regarding the intent labels used in our work**
>
> The original paper for SciCite provides two models. Here we only use the ACL-ARC model from that work, hence we only report the performance of 67.9. We chose the ACL-ARC label set due to its higher granularity, which we deemed desirable for our task. We will mention this in the main text of the paper to avoid confusion.
>
>
> -- **Regarding human validation for the citation intent and CORWA labelling**
>
> We adopt citation intent and CORWA schemata from prior work and rely on the performance measurements from prior work to judge their validity for our task. A systematic, non-anecdotal human evaluation of citation and CORWA performance would require substantial resources and lies beyond our scope. We will highlight this point in the limitation section.
>
>
> -- **Regarding inter-rater reliability**
>
> Originally 150 instances were annotated by three annotators (A, B, C) and each annotator annotated a distinct set of 50 instances. We conduct additional human annotations by randomly selecting 25 instances from each of the first two annotators' sets,  and then annotator C annotate these 50 instances. The Quadratic Cohen’s kappa is 0.48 (readability) score and 0.18 ( consistency) between the first annotator pair (A-C), as well as 0.52 (readability) and 0.42 (consistency) between the second annotator pair (B-C). The reason that the consistency agreement score is a bit lower for the first annotator pair A-C is that annotator C is a bit stricter with the consistency score than annotator A. We will report this inter-annotator reliability study in the final version.
>
>
> -- **Regarding the task definition**
>
> Indeed, we want to “generate a citation text” that is as close as possible to the original “$ T \in D^s $” – but the generated text does not itself belong to the document. The citing document contexts are *distinct* from the citation text e.g. “$ T \notin \{C_1,...,C_m\}  $”. We will clarify this in the camera-ready version.

---

### Official Review · Reviewer_cNNt · 2023-08-05

**Typos Grammar Style And Presentation Improvements:** N/A
**Soundness:** 3

**Excitement:**

3: Ambivalent: It has merits (e.g., it reports state-of-the-art results, the idea is nice), but there are key weaknesses (e.g., it describes incremental work), and it can significantly benefit from another round of revision. However, I won't object to accepting it if my co-reviewers champion it.

**Missing References:**

Mao, Y., Zhong, M., & Han, J. (2022, December). CiteSum: Citation Text-guided Scientific Extreme Summarization and Domain Adaptation with Limited Supervision. In Proceedings of the 2022 Conference on Empirical Methods in Natural Language Processing (pp. 10922-10935).

Wang, Y., Song, Y., Li, S., Cheng, C., Ju, W., Zhang, M., & Wang, S. (2022, June). DisenCite: Graph-Based Disentangled Representation Learning for Context-Specific Citation Generation. In Proceedings of the AAAI Conference on Artificial Intelligence (Vol. 36, No. 10, pp. 11449-11458).

**Paper Topic And Main Contributions:**

In this research, CITEBENCH is introduced as a comprehensive benchmark for citation text generation, consolidating diverse datasets and facilitating consistent evaluation of citation text generation models across various task designs and domains. By employing this novel benchmark, the authors thoroughly examine the performance of several robust baselines, assessing their adaptability across different datasets. Furthermore, the study provides valuable insights into task definition and evaluation methods, offering guidance for future research in the field of citation text generation.

This paper presents a comprehensive set of experiments conducted to evaluate the newly integrated benchmark.

**Questions For The Authors:**

A: What are the core distinctions that you perceive between your paper and previous works on citation generation datasets in this field?

B: Is it feasible to apply the general text generation benchmark population methods to this specific area?

C: What about exploring a broader range of state-of-the-art generation models for the citation text generation task?

**Reasons To Accept:**

1) This paper is well-organized, and easy to follow.

2) This paper is relevant to the topics of the conference.

3) It is meaningful to bridge NLP and bibliometrics.

**Reasons To Reject:**

The paper first redefines the task, and populates the four existing different existing benchmarks without introducing the detail of the task specific modification. The authors introduce a new task benchmark however the four origin benchmarks are from different tasks. How to populate the four different datasets to fit the new task formulation is not clearly introduced.

The problem formulation presented in this paper lacks novelty and does not show significant differences compared to other citation generation tasks, such as generating citations for related works.

Many existing studies are merely listed without undergoing in-depth analysis.

The findings appear to be trivial, and the paper would benefit from a more comprehensive discussion.

**Reproducibility:**

3: Could reproduce the results with some difficulty. The settings of parameters are underspecified or subjectively determined; the training/evaluation data are not widely available.

**Reviewer Confidence:**

5: Positive that my evaluation is correct. I read the paper very carefully and I am very familiar with related work.

---

> ### Author Rebuttal · Authors · 2023-08-28
>
> Thank you for taking your time and giving feedback on our work, we appreciate the detailed comments. Please see our responses to your comments below.
>
> **How the four datasets were transformed into the new task**
> - We are happy to clarify here and in the appendix. Each of the datasets was adapted to the new task definition by taking the different types of inputs e.g. *cited documents* and *abstract of citing paper* and concatenating these into a list that serves as input to the model along with special separators to distinguish the items, such as “<abs></abs>”, for the citing paper’s abstract. This transformation process will be included in the code that we will release along with the paper upon acceptance.
>
> **The problem formulation presented in this paper lacks novelty**
> - Our main goal in this paper is unification of the existing task definitions rather than designing a novel task definition. Our motivation is that there are a lot of papers that define the citation text generation task differently, and with all the different task definitions it becomes difficult to compare solutions and determine the SOTA. The unified task definition facilitates comparison between different approaches to citation text generation, which is important because in a real world scenario we want to know which solution would work best in a given setting.
> Furthermore, new datasets in this area can be easily integrated into our benchmark by extending the cited paper representation set D and the citing context set C (Line 173-176), our unified dataset structure allows to integrate new definitions that include additional textual information in the future, such as whole texts from the cited papers, or citation intents.
>
> **About the core distinctions of our paper and previous works on citation generation datasets in this field and more comprehensive discussion**
> - First, much of previous work has defined its own version of the task, created a dataset according to that task definition and designed a model to solve the task. This has led to a large number of highly divergent task setups and datasets, making it hard to compare approaches to citation text generation. Rather than focusing on creating a new task and dataset, here we examine and compare the current approaches. We also propose new evaluation metrics to look at the results from the perspectives of citation intent and discourse structure, which, as far as we know, has not been done in the field of citation text generation.
> - Second, writing a good related work section requires a very high level of expertise which requires a mental model of a big picture of the field. There is a wealth of literature on how to write a good literature review and on studying the elements/structures of different types of literature reviews [1] [2] [3]. In general, writing a literature review is closely related to writing a related work section. Unfortunately, the current task definitions in previous NLP studies on citation text generation overly simplify this very complex process, which hinders the development of successful real-world applications. In our work, the new knowledge we contribute to this field beyond a unified benchmark is discussed in Section 5: 1) we report the problematic​ ROUGE calculation practice in the previous studies on citation text generation; 2) more importantly, through our human evaluations, we identify a few problems of the existing datasets, and question the current definitions for the citation text generation task. In general, we view our work represents a crucial first step in approaching the extremely challenging problem of AI-assisted writing related work. We will make it more visible in the abstract and the introduction.
>
> **Is it feasible to apply the general text generation benchmark population methods to this specific area?**
> - In principle, general text generation benchmark population methods such as GEM v2 [4] can be applied to this specific area. The problem is that if each dataset owner contributes their dataset directly to the GEM benchmark using different task definitions, we still end up with a large number of highly divergent task setups and datasets. One of the core contributions of our work is to provide **a unified and standardized dataset structure** for citation text generation which allows researchers to compare different approaches easily.
>
> **What about exploring a broader range of state-of-the-art generation models for the citation text generation task?**
> - The current paper is a benchmarking paper which contributes the resources including dataset, evaluation metrics as well as solid baselines for other researchers to evaluate their models on. Similar to other benchmarking papers such as SQuAD [5], GLUE [6] and KILT [7], our benchmark and standardized evaluation enables systematic comparison of a wide range of generation models for citation text generation in the future.
>
> **About the missing references**
> - Thank you, we will add those to our related work overview.
>
> References:
>
> [1] JRandolph, Justus (2009) "A Guide to Writing the Dissertation Literature Review," Practical Assessment, Research, and Evaluation: Vol. 14, Article 13.
>
> [2] Diana Ridley. 2012. The Literature Review: A Step-by-Step Guide for Students. SAGE Publications.
>
> [3] Maria J. Grant and Andrew Booth. 2009. A typology of reviews: An analysis of 14 review types and associated methodologies. Health Information & Libraries Journal, 26(2):91–108.
>
> [4] Sebastian Gehrmann, Abhik Bhattacharjee, Abinaya Mahendiran, Alex Wang, Alexandros Papangelis, Aman Madaan, Angelina Mcmillan-major, Anna Shvets, Ashish Upadhyay, Bernd Bohnet, Bingsheng Yao, Bryan Wilie, Chandra Bhagavatula, Chaobin You, Craig Thomson, Cristina Garbacea, Dakuo Wang, Daniel Deutsch, Deyi Xiong, et al.. 2022. GEMv2: Multilingual NLG Benchmarking in a Single Line of Code. In Proceedings of the 2022 Conference on Empirical Methods in Natural Language Processing: System Demonstrations, pages 266–281, Abu Dhabi, UAE. Association for Computational Linguistics.
>
> [5] Pranav Rajpurkar, Jian Zhang, Konstantin Lopyrev, and Percy Liang. 2016. SQuAD: 100,000+ Questions for Machine Comprehension of Text. In Proceedings of the 2016 Conference on Empirical Methods in Natural Language Processing, pages 2383–2392, Austin, Texas. Association for Computational Linguistics
>
> [6] Alex Wang, Amanpreet Singh, Julian Michael, Felix Hill, Omer Levy, and Samuel Bowman. 2018. GLUE: A Multi-Task Benchmark and Analysis Platform for Natural Language Understanding. In Proceedings of the 2018 EMNLP Workshop BlackboxNLP: Analyzing and Interpreting Neural Networks for NLP, pages 353–355, Brussels, Belgium. Association for Computational Linguistics.
>
> [7] Fabio Petroni, Aleksandra Piktus, Angela Fan, Patrick Lewis, Majid Yazdani, Nicola De Cao, James Thorne, Yacine Jernite, Vladimir Karpukhin, Jean Maillard, Vassilis Plachouras, Tim Rocktäschel, and Sebastian Riedel. 2021. KILT: a Benchmark for Knowledge Intensive Language Tasks. In Proceedings of the 2021 Conference of the North American Chapter of the Association for Computational Linguistics: Human Language Technologies, pages 2523–2544, Online. Association for Computational Linguistics.

---

### Official Review · Reviewer_GP3Z · 2023-08-10

**Typos Grammar Style And Presentation Improvements:** 1. To improve the readability of the …
**Soundness:** 5

**Excitement:**

5: Transformative: This paper is likely to change its subfield or computational linguistics broadly. It should be considered for a best paper award. This paper changes the current understanding of some phenomenon, shows a widely held practice to be erroneous in someway, enables a promising direction of research for a (broad or narrow) topic, or creates an exciting new technique.

**Paper Topic And Main Contributions:**

# Paper Summary

The paper introduces a novel dataset for the task of Citation Text Generation of scientific documents. Its main contribution is a formalization of the task which allows the unification of several existing datasets and the standardization of the evaluation of the task. They also test several models from the state of the art on the resulting dataset and perform a comprehensive analysis of the results via qualitative and quantitative results.

# Contributions

1. A clever formalization of the task of Citation Text Generation that allows the unification of several existing resources and the standardization of the evaluation of the task.
2. A large-scale dataset for the task of Citation Text Generation of scientific documents.
3. An standardized evaluation methodology for the task based on quantitaive and qualitative metrics and human analysis.
4. A range of baseline implementations of models from the state of the art to foster research in this task.

**Reasons To Accept:**

1. This paper introduces a clever formalization of the task of Citation Text Generation that allows the unification of several existing resources and the standardization of the evaluation of the task.

2. The paper is very well written: it explains the task clearly, the figures are well-made and are very useful to understand the paper and the overall presentation is very clear.

3. The experiments in the paper are very solid: they described clearly the variability of their results, specified the training of the models completely, averaged their results over many runs, specified their computational budget and justified well the use of quantitative and qualitative tools.

**Reasons To Reject:**

I don't see any reason to reject this paper.

**Reproducibility:**

5: Could easily reproduce the results.

**Reviewer Confidence:**

4: Quite sure. I tried to check the important points carefully. It's unlikely, though conceivable, that I missed something that should affect my ratings.

---

> ### Author Rebuttal · Authors · 2023-08-28
>
> We would like to thank you for taking your time and are happy to hear that you appreciated our work! We will follow your suggestion to improve readability of the tables.
>
> Regarding the color scale in Figures 3 and 4, we opted for using different color scales since the magnitude of KL divergence differs between CORWA and citation intent classifiers.The values for citation intent are on average much lower than the values for CORWA classification. Our main goal here is to compare the differences between the models and not between CORWA and citation intents. Thus, if the same color scale were used for both plots, all the cells in Figure 3 would be close to white, making it difficult to visually compare the models on the citation intent dimension. We will think about how to highlight the scale difference in the camera-ready version.

---

### Meta-Review · Area_Chair_Nofm · 2023-09-17

**Recommendation:** 4

**Metareview:**

This paper presents CITEBENCH, a novel benchmark for citation text generation. The authors propose an approach for merging and harmonizing diverse datasets, ensuring a consistent evaluation of models across different tasks and domains. The work also includes an exhaustive examination of several robust baselines, offering insights into their adaptability across different datasets. The paper is well-received with some minor concerns raised by the reviewers. The reviewers praised the novelty of the approach, the clarity of presentation, and the thoroughness of the experiments described. The reviewers also highlighted the potential of this work to become a reference benchmark in the field of Citation Text Generation.

The paper introduces a novel dataset for Citation Text Generation and provides a clever formalization of the task that allows the unification of several existing resources and the standardization of the evaluation of the task. The paper contributes significantly to the field by providing an exhaustive set of baselines for future work, a standardized evaluation methodology, and a comprehensive analysis of the results. The quality of the paper is high, with clear explanations, well-made figures, and solid experiments. On the other hand, one reviewer raised concerns about the clarity of task-specific modifications when populating the new benchmark from different existing benchmarks. There were questions about the novelty of the problem formulation and a perceived lack of depth in the analysis of existing studies, which was tried to answer in rebuttal. A minor weakness identified by a reviewer is the absence of validation done to check if the labeling of the benchmark or the generated texts actually reflects how humans would label them.

Given the novelty and potential impact of the work, as well as the overall positive feedback from the reviewers, the paper is recommended for acceptance. However, the authors are strongly encouraged to address the reviewers' concerns and suggestions in the final version of the paper.

---

### Decision · Program_Chairs · 2023-10-07

**Decision:**

Accept-Main

**Comment:**

This paper presents CITEBENCH, a novel benchmark for citation text generation. The authors propose an approach for merging and harmonizing diverse datasets, ensuring a consistent evaluation of models across different tasks and domains. The work also includes an exhaustive examination of several robust baselines, offering insights into their adaptability across different datasets. The paper is well-received with some minor concerns raised by the reviewers. The reviewers praised the novelty of the approach, the clarity of presentation, and the thoroughness of the experiments described. The reviewers also highlighted the potential of this work to become a reference benchmark in the field of Citation Text Generation.

The paper introduces a novel dataset for Citation Text Generation and provides a clever formalization of the task that allows the unification of several existing resources and the standardization of the evaluation of the task. The paper contributes significantly to the field by providing an exhaustive set of baselines for future work, a standardized evaluation methodology, and a comprehensive analysis of the results. The quality of the paper is high, with clear explanations, well-made figures, and solid experiments. On the other hand, one reviewer raised concerns about the clarity of task-specific modifications when populating the new benchmark from different existing benchmarks. There were questions about the novelty of the problem formulation and a perceived lack of depth in the analysis of existing studies, which was tried to answer in rebuttal. A minor weakness identified by a reviewer is the absence of validation done to check if the labeling of the benchmark or the generated texts actually reflects how humans would label them.

Given the novelty and potential impact of the work, as well as the overall positive feedback from the reviewers, the paper is recommended for acceptance. However, the authors are strongly encouraged to address the reviewers' concerns and suggestions in the final version of the paper.